# The Impact of Bamboo Consumption on the Spread of Antibiotic Resistance Genes in Giant Pandas

**DOI:** 10.3390/vetsci10110630

**Published:** 2023-10-24

**Authors:** Zheng Yan, Xin He, James Ayala, Qin Xu, Xiaoqiang Yu, Rong Hou, Ying Yao, He Huang, Hairui Wang

**Affiliations:** 1Chengdu Research Base of Giant Panda Breeding, Chengdu 610081, China; 202131200032@mail.bnu.edu.cn (Z.Y.); giantpanda721@gmail.com (J.A.); xuqinpanda@panda.org.cn (Q.X.); danielle.ll@foxmail.com (X.Y.); hourong@panda.org.cn (R.H.); yaoying2018@panda.org.cn (Y.Y.); youngtree@163.com (H.H.); 2Sichuan Key Laboratory of Conservation Biology for Endangered Wildlife, Chengdu 610081, China; 3Sichuan Academy of Giant Panda, Chengdu 610081, China; 4Key Laboratory for Biodiversity and Ecological Engineering of Ministry of Education, Department of Ecology, College of Life Sciences, Beijing Normal University, Beijing 100875, China

**Keywords:** giant panda, environment, pollution, bamboo consumption, antibiotic resistance genes, gut microbiota

## Abstract

**Simple Summary:**

The focus of this study was to explore the impact of the transmission of antibiotic resistance genes (ARGs) from a bamboo diet to the gut microbiota of giant pandas and further dissemination through fecal matter. Metagenomic analysis and the Comprehensive Antibiotic Resistance Database (CARD) were used to investigate the presence of ARGs in the gut microbiota of captive giant pandas during the consumption of different parts of bamboo. The results showed that the number of ARGs was highest in gut microbiota of the giant panda during the consumption of bamboo leaves, while the variety of ARGs was highest during the consumption of shoots. Specific bacteria associated with ARG dissemination were also identified. This study emphasizes the need for the proper handling of panda feces and regular monitoring of antimicrobial-resistant genes to mitigate the threat of antibiotic resistance. Overall, this research provides insights into the complex relationship between diet, gut microbiota, and health in giant pandas, contributing to their conservation and management.

**Abstract:**

The spread of antibiotic resistance genes (ARGs) in the environment exacerbates the contamination of these genes; therefore, the role plants play in the transmission of resistance genes in the food chain requires further research. Giant pandas consume different bamboo parts at different times, which provides the possibility of investigating how a single food source can affect the variation in the spread of ARGs. In this study, metagenomic analysis and the Comprehensive Antibiotic Resistance Database (CARD) database were used to annotate ARGs and the differences in gut microbiota ARGs during the consumption of bamboo shoots, leaves, and culms by captive giant pandas. These ARGs were then compared to investigate the impact of bamboo part consumption on the spread of ARGs. The results showed that the number of ARGs in the gut microbiota of the subjects was highest during the consumption of bamboo leaves, while the variety of ARGs was highest during the consumption of shoots. Escherichia coli, which poses a higher risk of ARG dissemination, was significantly higher in the leaf group, while *Klebsiella*, *Enterobacter*, and *Raoultella* were significantly higher in the shoot group. The ARG risk brought by bamboo shoots and leaves may originate from soil and environmental pollution. It is recommended to handle the feces of giant pandas properly and regularly monitor the antimicrobial and virulence genes in their gut microbiota to mitigate the threat of antibiotic resistance.

## 1. Introduction

The prevalence of gastrointestinal disorders caused by pathogenic bacteria poses a major threat to the well-being of giant pandas [1,2,3]. In recent decades, the use of antimicrobial drugs has become a common practice in safeguarding the health of captive giant pandas against infectious diseases [1,4,5]. However, the widespread application of antimicrobial medications has resulted in the proliferation of antibiotic-resistant bacteria (ARB) and hastened the emergence and dissemination of multi-drug-resistant (MDR) strains in both humans and the surrounding ecosystem [6]. Some microorganisms carry genes that confer a specific resistance to antibiotics due to gene mutations [7]. Currently, the Comprehensive Antibiotic Resistance Database (CARD, https://card.mcmaster.ca/home, accessed on 4 August 2023) contains 5159 reference sequences of resistance genes. These resistance genes are mainly acquired from endogenous and exogenous sources [8,9]. Endogenous antibiotic resistance genes (ARGs) include values derived from the environment and microorganisms [9]. Exogenous ARGs primarily come from parallel movement [10]. Frequent transfers of giant pandas to other enclosures may lead to the spread of antimicrobial drug resistance [1]. For example, multidrug-resistant *Klebsiella pneumoniae* presents a serious menace to the health of giant pandas and potential risks to wildlife center visitors and giant panda caretakers [11]. Comparatively, the microbiota of captive giant pandas exhibits a higher prevalence of both virulence genes and antimicrobial resistance genes (ARGs) in contrast to their counterparts in the wild [12]. Continuous monitoring of antimicrobial drug resistance (AMR) in bacterial isolates from giant pandas is crucial for their protection and public health. Research has revealed that ARGs have the potential to be transmitted to the plant microbiome via soil, water, and the atmosphere; this leads to an elevation in the frequency of resistance genes within the surrounding environment [13,14,15]. Consequently, plants can serve as reservoirs and carriers of ARGs, directly contributing to their introduction into the host organism [16,17]. Furthermore, significant variations in gut ARGs have been observed in individuals with different dietary habits, including omnivores, ovo-lacto vegetarians, and vegans [18]. The escalation in the variety and amount of ARGs in food presents substantial safety hazards to the well-being of both humans and animals.

Research has indicated that ARGs are enriched in the microbiomes of captive giant pandas [19]. The accumulation of ARGs in different parts of plants varies, indicating that the species and abundance of ARGs are higher in the rhizoplane and leaf compared to other parts [20,21]. However, studying the mechanism of resistance gene transfer from different parts of plants to the host is challenging. Giant pandas, as highly specialized animals, exclusively feed on specific parts of bamboo during certain periods, providing a perfect model to study resistance genes from a single plant source [22]. Previous research has already discovered differences in blood metabolites [23], gut microbiota composition and function [24], fecal metabolites [25], and fiber digestion efficiency [26] when giant pandas consume different bamboo parts. The objective of this study was to investigate the characteristics of ARGs and the microbiome in the feces of giant pandas when they consumed different parts of bamboo, including the shoot, culm, and leaf, and explored the antibiotic resistance mechanisms and the interaction between ARGs and host microbiota. Understanding the species and characteristics of ARGs and microbiota in the feces of giant pandas will aid in the conservation and management of this vulnerable species. Furthermore, this research will contribute to enhancing our understanding of the transfer of ARGs from plants to animals, and the potential risks posed to both animal and human health.

The objective of this study was to examine the microbiota and ARGs in giant panda feces under varying bamboo consumption conditions.

## 2. Materials and Methods

### 2.1. Experimental Design and Data Source

We obtained nine metagenomic raw data from the gut microbiota of eight captive giant pandas (five males and three females, aged 8 to 13) provided with three different bamboo parts diets: bamboo culm, bamboo leaves, and bamboo shoots. These data were acquired from our previous study [24], where each diet phase consisted of three samples. Notably, prior to metagenomic testing, the gut microbiota of 19 giant pandas at three different feeding stages was analyzed by sequencing the 16S rRNA of 27 fecal samples (nine samples per group). The metagenomic samples used in this study were selected based on the sequencing results of the 16S rRNA sequencing, employing the representative community selection method, which is currently the most commonly used approach for screening metagenomic samples [27]. Therefore, the metagenomic data for each stage can represent the gut microbiota information of giant pandas during the feeding stages of these three bamboo parts.

All samples were collected at the Chengdu Research Base of Giant Panda Breeding, Sichuan, China. All giant pandas were healthy and had not received any antibiotic treatment during the sample collection phase. Additionally, there were no occurrences of estrus, pregnancy, or nursing activities among the female giant pandas during the research period.

### 2.2. Bioinformatics Analysis

Due to updates in databases and analysis methods, we reanalyzed the raw sequences to obtain more useful data and enhance the credibility of the research results. The Readfq tool (https://github.com/cjfields/readfq, accessed on 5 August 2023) was utilized for preprocessing the raw data obtained from the Illumina sequencing platform, resulting in clean data for subsequent analysis. To address the potential contamination from host sequences, the clean data was subjected to BLAST against the host database using Bowtie2 software (V.2.2.4, http://bowtie-bio.sourceforge.net/bowtie2/index.shtml, accessed on 5 August 2023) [28] to filter out reads that may originate from the host. The clean data were then subjected to assembly analysis using MEGAHIT software (v.1.0.4) [29], and scaffolds without N junctions were obtained by breaking the resulting scaffolds.

MetaGeneMark (v.2.10, http://topaz.gatech.edu/GeneMark/, accessed on 6 August 2023) was utilized to predict Open Reading Frames (ORFs) for Scaftigs (≥500 bp) in each sample [30]. ORFs with a length less than 100 nt were filtered out from the prediction results [31]. The resulting ORF prediction data underwent redundancy elimination using CD-HIT software (v. 4.5.8, http://www.bioinformatics.org/cd-hit/, accessed on 6 August 2023) to obtain a nonredundant initial gene catalog [32]. The clean data from each sample were aligned to the initial gene catalog using Bowtie2 to calculate the read counts for each gene in the sample alignments [33]. To obtain the final gene catalog (Unigenes) for further analysis, genes with read counts ≤ 2 in each sample were filtered out [34]. To analyze the differential expression of Unigenes across the three sample groups, ANOVA and Tukey’s HSD methods were employed. The Benjamini–Hochberg (BH) correction method was employed to control the False Discovery Rate (FDR) in the analysis. Diamond software (v.2.0.11, https://github.com/bbuchfink/diamond/, accessed on 6 August 2023) [35] was utilized to perform sequence alignment of the Unigenes with bacterial, fungal, archaeal, and viral genomes. These reference sequences were extracted from the NCBI NR database (Version: 2023.03, https://www.ncbi.nlm.nih.gov, accessed on 6 August 2023) [36]. In order to investigate the similarities between various samples, clustering analysis using Bray–Curtis distance was performed to construct a sample clustering tree. To identify significant differential species biomarkers between different sample groups, Linear discriminant analysis Effect Size (LEfSe) analysis was conducted. Differential abundance of species between different groups was evaluated using a rank sum test, and a linear discriminant analysis (LDA) was conducted for dimensionality reduction and to measure the effect size of differential species [37]. Evolutionary branching diagrams and LDA value distribution plots were visualized using species with an LDA score > 4.

### 2.3. Annotations of Resistance Gene

The unigenes were matched against the CARD database (v.3.2.6, https://card.mcmaster.ca/, accessed on 7 August 2023) [38] using the Resistance Gene Identifier (RGI) software (v.6.0.2) [39]. The CARD database was built upon the Antibiotic Resistance Ontology (ARO), which integrates information on ARGs, drug classes, antimicrobial resistance gene families (ARGM), resistance mechanisms (RM), and their relationships [39]. Based on the RGI alignment results and unigenes’ abundance information, the relative abundance (ppm, parts per million) of each ARG was calculated. Using the abundance of ARG, various analyses were performed, including abundance histograms, abundance clustering heatmaps, abundance distribution circle maps, ARG difference analysis between groups, identification of resistance genes (unigenes annotated as ARG), and species attribution analysis of resistance mechanisms. ANOVA and Tukey’s HSD methods were employed to analyze the significant differences in the quantity of resistance genes and ARGs among the three sample groups, with the False Discovery Rate (FDR) controlled using the Benjamini–Hochberg correction. The relative abundance of annotated ARGs was calculated for the generation of circle maps.

### 2.4. Differences in Resistance Genes among Gut Microbiota

Through the analysis of species annotations for all samples within each group, we obtained the respective species information associated with each resistance gene in the samples. Firstly, by comparing the number and type of ARGs attributed to gut microbiota species, a Sankey diagram was created to visually represent the presence of resistance gene information in the species. Then, using the Resistance Mechanism classification in the CARD database, a distribution diagram of resistance mechanisms was generated based on the relationship between the mechanisms of action of these resistance genes and the species. Finally, the gut microbiota with the highest contribution to resistance genes at various taxonomic levels were selected, and significant differences between groups were analyzed using ANOVA, with FDR controlled using BH correction, to identify the gut microbiota with the highest contribution to resistance genes associated with the consumption of different parts of bamboo by giant pandas.

## 3. Results

### 3.1. Microbial Annotation Information

Metagenomic shotgun sequencing of the macrogenome data resulted in 519,857 gene catalogues (unigenes) after assembly, filtering, and redundancy removal, which is more than double of the previous analysis (232,096). The abundance of gut microbiota unigenes in giant pandas was significantly higher when feeding on a bamboo shoot compared to a leaf and culm (Figure 1A). Additionally, the bamboo shoot group had the highest number of unique unigenes, while the bamboo culm group had the lowest (Figure 1B). All unigenes were annotated to 75 phyla, 1024 genera, and 3922 species-level microorganisms. The gut microbiota composition of giant pandas when consuming different parts of bamboo formed separate clusters on a dendrogram (Figure 1C). The gut microbiota biomarkers in the shoot group were mainly concentrated in Enterobacteriaceae and Lactobacillales. The biomarker for the leaf group was *Escherichia coli*. The biomarkers for the culm group were Aurantimonadaceae and Streptococcaceae (Figure 1D and Appendix A).

### 3.2. Abundance of Antibiotic Resistance Genes

After annotation with the CARD database, it was found that the gut microbiota of giant pandas feeding on bamboo leaves had the highest number of ARGs, but the differences among the three groups were not statistically significant (Figure 2A). Furthermore, a total of 137 antibiotic resistance genes (ARGs) were identified, with 130 ARGs in the shoot group, 104 ARGs in the leaf group, and 88 ARGs in the culm group. Among these, 81 ARGs were shared among the three groups, while 28 ARGs were unique to the shoot group, 4 ARGs were unique to the leaf group, and 1 ARG was unique to the culm group (Figure 2B). Each group exhibited unique ARG compositions, with the shoot group having over 60% abundance of *E. coli* EF-Tu mutants conferring resistance to Pulvomycin. The leaf group showed a more balanced distribution of ARGs, including *E. coli* EF-Tu mutants conferring resistance to Pulvomycin, tetA, mdtN, *E. coli* soxS with mutation conferring antibiotic resistance, cpxA, bacA, and EC-14. The culm group had three main ARGs, namely vanT gene in vanG cluster, vanY gene in vanB cluster, and qacG. The 137 ARGs were classified into 47 drug classes, with glycopeptide antibiotic, fluoroquinolone antibiotic, tetracycline antibiotic, macrolide antibiotic, and cephalosporin being the most prominent classes in terms of proportion.

### 3.3. Distribution of Gut Microbiota Resistance Gene

The gut microbiota of giant pandas exhibited the presence of 3 phyla, 27 genera, and 53 species in the annotation of resistance genes. At the phylum level, Pseudomonadota (also known as Proteobacteria) and Bacillota (also known as Firmicutes) were predominant, with 74 and 17 ARGs annotated, respectively. In terms of family-level classification, Enterobacteriaceae and Streptococcaceae were prominent, with 58 and 8 ARGs annotated, respectively. In addition, the less abundant families such as Clostridiaceae and others were annotated with seven ARGs. Among the genera, *Streptococcus* had the highest abundance and served as a major source of Bacillota resistance genes, with six ARGs annotated. *Escherichia* followed with 3 ARGs annotated, while *Enterobacter* had annotations for 18 ARGs, *Raoultella* for 4 ARGs, and *Clostridium* for 7 ARGs. At the species level, *Streptococcus alactolyticus* had the highest abundance, annotated with one ARG (qacG), and was followed by *E. coli* with one ARG annotated (mdtM) (Figure 3).

Furthermore, the attribution of ARGs in the gut microbiota of captive giant pandas can be categorized into 10 different resistance mechanisms (RMs). The most predominant RMs include antibiotic efflux, antibiotic inactivation, and antibiotic target alteration, which collectively account for an average relative abundance of over 70% and are primarily attributed to the phylum Pseudomonadota. A smaller portion of these RMs is attributed to the phylum Bacillota (Figure 4).

The relative proportion of resistance genes connected with each resistance mechanism and the gut microbiota is illustrated by the outer circle.

### 3.4. Differences in Gut Microbiota Composition Associated with ARGs

To analyze the differences in gut microbiota composition associated with annotated ARGs among different groups, we performed statistical analysis using the most important microbial resistance gene counts at each taxonomic level. At the phylum level, the number of annotated resistance genes in Pseudomonadota was significantly higher during the bamboo leaf consumption period compared to the shoot and culm consumption periods. Bacillota showed significantly higher resistance gene counts in the culm group compared to the other two groups. At the family level, Enterobacteriaceae dominated in the bamboo leaf group, being the major component of Pseudomonadota. Streptococcaceae showed significantly higher counts in the culm group compared to the other two groups. Among the genera, *Streptococcus*, which is the predominant member of Bacillota, exhibited a significantly higher abundance in the culm group compared to the other two groups. *Escherichia* showed the significantly higher counts in the bamboo leaf group. *Enterobacter* and *Klebsiella* exhibited the highest counts in the shoot group. However, there were no significant differences observed in the levels of annotated resistance genes associated with Clostridiaceae and *Clostridium* among three groups, which had relatively higher abundances. At the species level, *E. coli* was the most important component of *Escherichia* and showed significantly higher counts in the bamboo leaf group. *Streptococcus alactolyticus* exhibited the highest counts in the culm group (Figure 5).

## 4. Discussion

This study utilized updated analysis software and databases, allowing for a larger acquisition of effective gut microbiota gene data. This provides more robust evidence to investigate the impact of different nutritional components in the diet of giant pandas. Based on the species annotation from metagenomic analysis, it was found that the gut microbiota gene count was highest during the bamboo shoot consumption period, which is consistent with previous studies [24]. This may be attributed to the higher protein and fat content in bamboo shoots, providing a rich nutritional foundation for the microbiota [23]. Furthermore, it was observed that *E. coli* was identified as a biomarker in the gut microbiota of giant pandas during the bamboo leaf consumption period, while biomarkers such as *Klebsiella* and *Enterobacter* were associated with the bamboo shoot consumption period. These bacteria are commonly found in the gut of humans and other animals, including giant pandas, and play important roles in symbiosis [40,41]. However, they can also be opportunistic pathogens and cause various diseases [42]. The significantly higher presence of *Streptococcus* in the bamboo culm group is noteworthy, as *Streptococcus* are the first inhabitants of the oral cavity and play important roles as symbiotic bacteria [43]. However, they can also be opportunistic pathogens and cause systemic infections such as infective endocarditis [43]. Abundant *Streptococcus* has also been detected in the oral cavity of giant pandas [44,45]. It is speculated that the higher abundance of *Streptococcus* in the gut microbiota during the bamboo culm consumption period may be partially attributed to the entry of oral *Streptococcus* into the gut through increased chewing of bamboo culms. *Streptococcus* in the gut of giant pandas is often classified as probiotic [46]. Therefore, based on the gut microbiota annotation, it can be inferred that giant pandas face a lower risk of pathogenic bacteria during the bamboo culm consumption stage.

Resistance genes can indeed move and accumulate in the air, water, soil, and plants through genetic elements [16,47], ultimately entering the host body through food [48,49]. This poses a potential threat to host health and epidemic prevention. In this study, through the annotation of ARGs, it was found that the number of ARGs was highest during the leaf feeding stage, and the variety of ARGs was the largest in the feces of giant pandas during the shoot feeding stage. Bamboo shoots have intensive contact with soil, leading to the enrichment of a large number of ARGs [50]. Some of these resistance genes may transfer to leaves and accumulate in large quantities [51]. Captive pandas consume different parts of bamboo during different periods, with leaf feeding being more common in winter [23]. Environmental pollution such as PM_10_, PM_2.5_, SO_2_, and O_3_ in Chengdu, China is also more severe during winter than in other seasons [52,53], which may contribute to the intake of more ARGs during this period. Other studies have also found a higher trend of ARGs in December compared to July in *E. coli* isolates [1]. The source of ARGs in the feces of giant pandas may be attributed to transportation and the captive environment [54]. This suggests that attention should be paid to the captive environment and food conditions of giant pandas. Through the analysis of ARGs in giant panda feces, it was found that a variety of antibiotic types were present, indicating that the source of resistance is complex, and that environmental pollution in captive giant pandas may be significant [12].

The dominant resistance genes in the study were derived from Pseudomonadota and predominantly annotated to Enterobacteriaceae, both of which were significantly higher in the bamboo leaf group. It is worth noting that many pathogenic bacteria, such as *Escherichia*, *Klebsiella*, *Enterobacter*, and *Raoultella*, belong to this taxonomic group [11,55,56], posing a serious threat to the survival of giant pandas. *Klebsiella* and *Enterobacter* can cause a wide range of infections [55], and their antibiotic resistance has been a major concern [56]. *Escherichia* and *E. coli*, which were annotated with resistance genes, were also significantly higher in the bamboo leaf group. Previous research has confirmed the widespread presence of antimicrobial and disinfectant resistance in *E. coli*, and its resistance is related to animal habitats [1]. Therefore, *E. coli* can serve as an important medium for the spread of disinfectant and antimicrobial resistance. Investigations on the prevalence of antimicrobial and disinfectant resistance genes in *E coli* and *K. pneumoniae* samples from giant pandas have shown that *E. coli* infections are often caused by multidrug-resistant strains [3,5]. The presence of ARGs and virulence-associated genes (VAGs) in *E. coli* is a growing concern as they are considered environmental pollutants with detrimental impacts on human and animal health [57].

The annotation of resistance genes for *Klebsiella*, *Enterobacter*, and *Raoultella* were significantly higher in the bamboo shoot group. Specifically, *Enterobacter* species may exhibit a natural resistance to ampicillin, kanamycin, and tetracycline [58]. Antibiotic-resistant *E. coli* strains derived from captive giant pandas are a combination of ARGs, VAGs, and mobile gene elements (MGEs), and there is a significant correlation among them [19]. The considerable prevalence of resistance genes and the significant positive correlation observed between resistance genes and VAGs underline the importance of continuous monitoring to assess the impact of antibiotic resistant *E. coli* transmission [59]. *Raoultella*, like *Klebsiella*, is recognized as a potentially significant pathogen due to its ability to acquire antimicrobial resistance genes (ARGs) from other bacteria. This acquisition contributes to the rise of multidrug-resistant strains [60]. Additionally, *Raoultella* shares similar virulence factors with *Klebsiella*. [60]. *Klebsiella* and *Raoultella* are commonly found Gram-negative facultative anaerobic rods in nature, such as in water and soil [61,62]. Therefore, some of the resistance genes in bamboo shoots may originate from the local soil and water sources.

In addition, this study found that the number of resistance genes annotated to *K. pneumoniae* were significantly higher in the shoot and leaf groups compared to the culm group. *K. pneumoniae* is naturally resistant to certain antibiotics, including ampicillin, amoxicillin, carbenicillin, and ticarcillin [63]. Previous research has isolated multi-drug-resistant *K. pneumoniae* from fecal samples of captive giant pandas, which carry a large number of ARGs [64]. Multi-drug-resistant *K. pneumoniae* are resistant to three or more classes of antibiotics [65,66]. Hemorrhagic enteritis has been observed in giant pandas infected with *K. pneumoniae*, which has further progressed to septicemia. This serious condition has led to genitourinary hematuria and ultimately the death of the affected pandas [67]. Infections attributable to *K. pneumoniae* have shown a rise in captive populations of giant pandas [68,69]. These infections often manifest as mixed infections with other bacteria and have emerged as a significant pathogen among captive giant pandas [64]. Therefore, during the consumption of bamboo leaves and shoots, giant pandas are at a higher risk of pathogenic intestinal microbiota, and the phenomenon of bacterial resistance is more severe. It is important to properly handle the feces from bamboo leaves and shoots to prevent the spread of antibiotic resistance and maintain public health and safety. To prevent the threat of ARGs to captive giant pandas and humans, regular monitoring of the diversity of antimicrobial-resistant and virulence genes in their gut microbiota is recommended.

## 5. Conclusions

In conclusion, this study utilized advanced analysis techniques and databases to investigate the impact of different nutritional components in the diet of giant pandas on their gut microbiota and ARGs. The consumption of bamboo leaves and shoots by giant pandas has implications for their gut microbiota composition and the presence of ARGs. The higher abundance of resistance genes during the leaf and shoot consumption stages can be attributed to factors such as soil pollution and environmental contamination. These findings highlight the importance of proper handling of bamboo leaf and shoot feces to prevent the spread of antibiotic resistance and maintain public health. Regular monitoring of antimicrobial-resistant and virulence genes in the gut microbiota of captive giant pandas is recommended to mitigate the threat of antibiotic resistance. Overall, this research contributes to our understanding of the complex relationship between diet, gut microbiota, and health in giant pandas, and provides valuable insights for their conservation and management as an endangered species.

## Figures and Tables

**Figure 1 vetsci-10-00630-f001:**
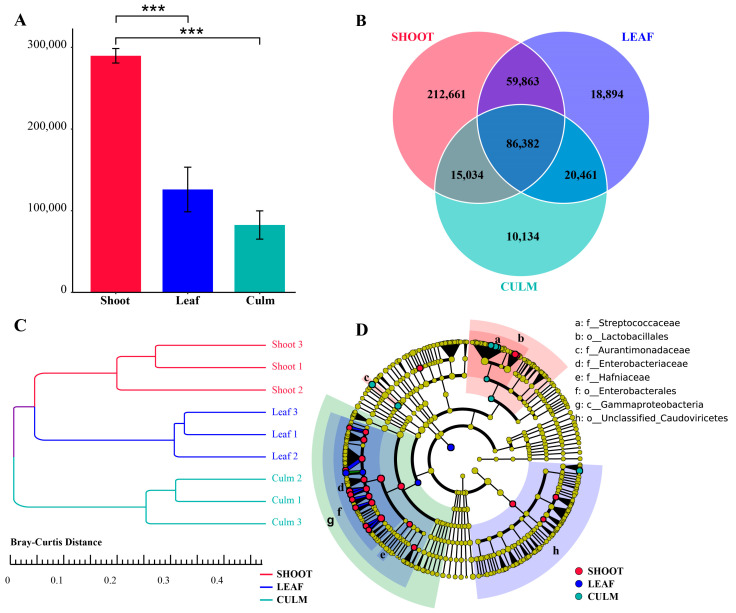
Gut microbiota gene annotation and composition differences in giant pandas feeding on different parts of food. (**A**) Bar plot showing the differential abundance of unigenes among the three groups, *** *p* < 0.001. (**B**) A Venn diagram is employed to visually represent the distinct and overlapping unigenes across the various groups. (**C**) Clustering tree based on Bray-Curtis distance. (**D**) Phylogenetic tree of differentially abundant species. The concentric circles depict the taxonomic hierarchy, ranging from phylum to species. Each individual circle at each taxonomic level represents a classification at that level, with the size of the circle reflecting its relative abundance. Species that do not exhibit significant differences are denoted by yellow coloration, while differentially abundant species biomarkers are color-coded based on their respective groups, indicated by an LDA score > 4.

**Figure 2 vetsci-10-00630-f002:**
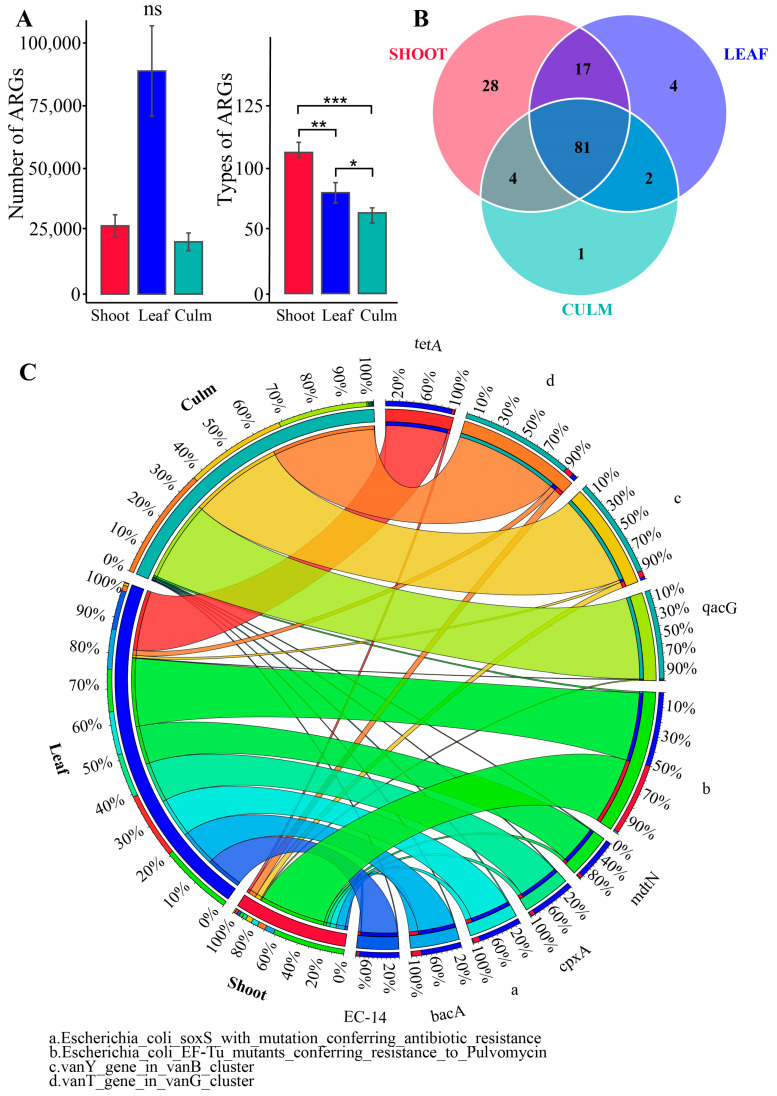
The annotation of antibiotic resistance genes (ARGs) was performed on the gut microbiota of giant pandas. (**A**) Bar plot showing the inter-group differences in number and type of ARGs. * *p* adjust < 0.05, ** *p* < 0.01, *** *p* < 0.001, ns, not significant. (**B**) Venn diagram is employed to visually represent unique and shared ARG counts among the three groups. (**C**) Circular plot depicting the overview of resistance genes, displaying the relative abundance of the top ten ranked ARGs in each group. The abundance of antibiotic resistance genes (ARGs) in different groups is represented on the left, while the right side displays the relative abundance of each ARG in the samples.

**Figure 3 vetsci-10-00630-f003:**
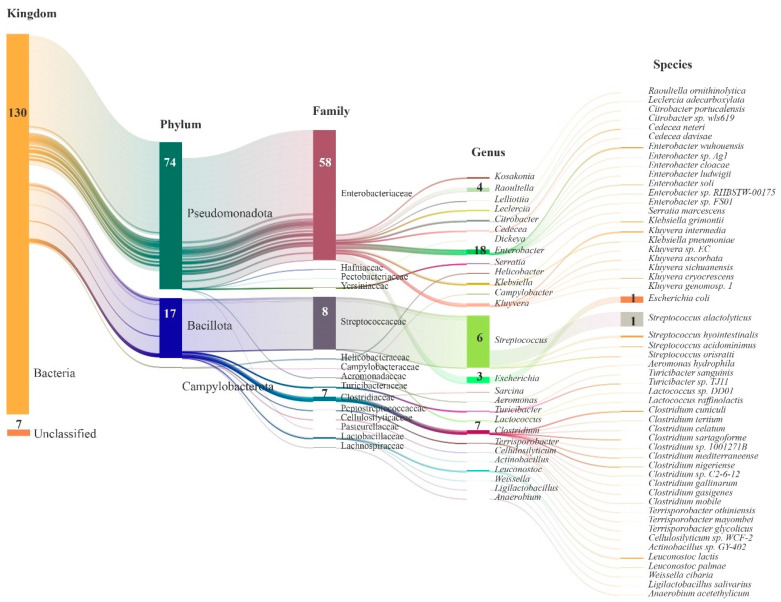
Sankey diagram illustrating the taxonomic classification of species identified with ARGs in the gut microbiota of giant pandas. The length of each column represents the relative abundance of the respective microbial group. The numbers within the columns indicate the count of ARGs.

**Figure 4 vetsci-10-00630-f004:**
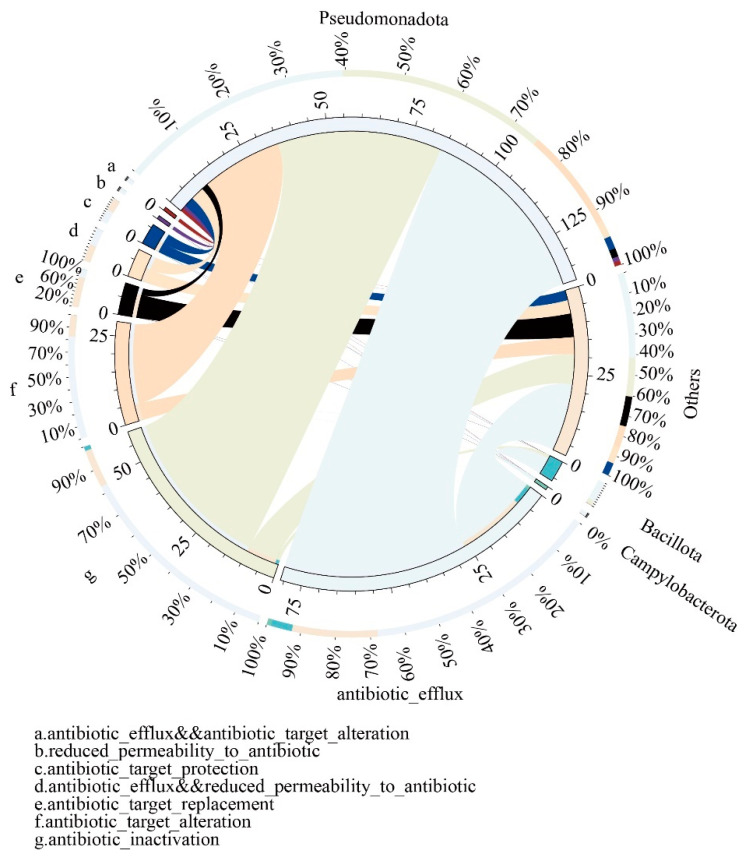
Overview circular plot depicting the relationship between resistance mechanisms and gut microbiota. The inner circle represents different species and their corresponding resistance mechanisms (RMs), indicated by different colors. The scale represents the number of RMs. The outer circle represents the relative proportion of resistance genes associated with each resistance mechanism and the gut microbiota.

**Figure 5 vetsci-10-00630-f005:**
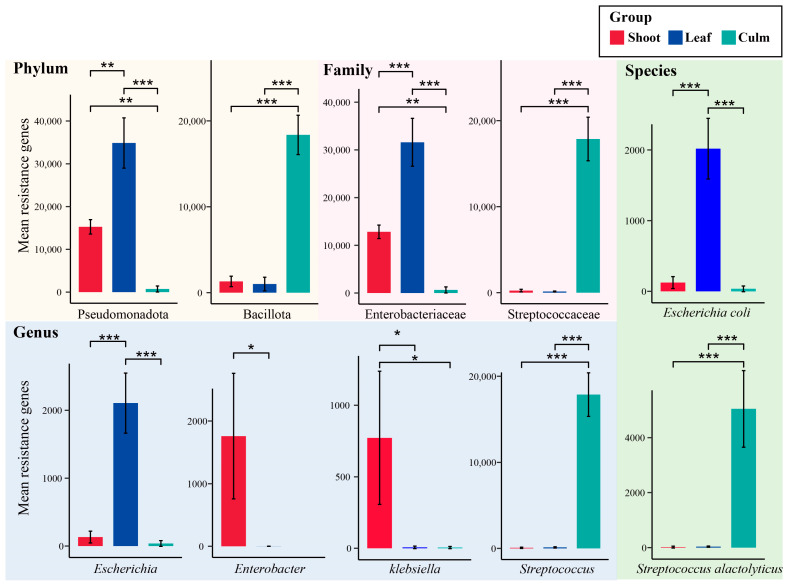
Important gut microbiota species with significant differences in annotated antibiotic resistance genes (ARGs). The background color represents the taxonomic hierarchy of gut microbiota. * *p* adjust < 0.05, ** *p* adjust < 0.01, *** *p* adjust < 0.001.

## Data Availability

The clean data, following the removal of host sequences and quality control, has been successfully uploaded to the Sequence Read Archive (SRA) under the BioProject PRJNA1004900.

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
