# Peer review of "The Impact of Bamboo Consumption on the Spread of Antibiotic Resistance Genes in Giant Pandas"

_vetsci, 2023, doi:10.3390/vetsci10110630_

Round 1

Reviewer 1 Report

In this manuscript, the authors investigate the effect of consumption of different parts of the bamboo plant on gut microbiome composition and presence of antibiotic resistance genes in Giant panda. Metagenomic analysis and the Comprehensive Antibiotic Resistance Database (CARD) were used to identify and classify antibiotic resistance genes.

Overall, the manuscript is well written and the conclusions are supported by the results.

Following are some minor comments to enhance the strength of the manuscript

Line52-53: Authors should clarify that ARGs are transmitted to plant microbiome and not the plant cells.

Line 105: Please define ARO in the text before using the abbreviation. Did the authors mean “ARG” were identified?

Author Response

Dear Reviewer,

Thank you for your valuable feedback on our manuscript. We have taken your advice into consideration and made the necessary updates to improve the clarity of our research findings. Specifically, we have added additional information to clarify the transmission of ARGs to the plant microbiome. Furthermore, we have replaced three references with more suitable ones in that section to enhance the credibility of our study.

We have carefully considered your suggestion regarding the use of the term "ARO" in our article, and we agree that it may be confusing for readers who are not familiar with ontologies. Therefore, we have made revisions to the manuscript to replace "ARO" with "ARGs" consistently throughout the text.

I appreciate your comments and believe that our revised manuscript is now clearer and more concise. Thank you again for your time and efforts in reviewing our work.

Sincerely,

Hairui Wang

Reviewer 2 Report

This original paper delves into a novel and intriguing study that examines the spread of antibiotic resistance genes (ARGs) in the context of giant panda dietary habits. The paper explores how the consumption of different parts of bamboo by giant pandas impacts the prevalence and variety of ARGs in their gut microbiota. This study employs metagenomic analysis and the Comprehensive Antibiotic Resistance Database (CARD) to annotate ARGs and investigate their variations based on bamboo consumption. The findings provide valuable insights into the impact of panda diets on ARGs and offer recommendations for managing this issue. The study's approach is innovative and timely, as antibiotic resistance is a global concern. Investigating the role of giant pandas in this context is a fresh and intriguing perspective that has not been extensively explored. This study can set the stage for further research in similar areas, possibly in other species with distinct diets. The paper rightly concludes with a recommendation to manage giant panda feces properly and monitor antimicrobial and virulence genes in their gut microbiota. However, it would be valuable to discuss the practical implications and challenges of implementing these recommendations. In summary, the paper is a valuable contribution to the field of antibiotic resistance research, shedding light on an unconventional vector for ARG transmission. It offers a sound methodology and intriguing results. Overall I had a great pleasure reading this manuscript, as it presents a novel point of view and is very carefully prepared.

Author Response

Dear Reviewer,

Thank you for your positive and valuable feedback on my original research manuscript. I am glad that you found the study to be innovative and timely, addressing a significant global concern of antibiotic resistance. The investigation of the role of giant pandas in the transmission of antibiotic resistance genes (ARGs) in the context of their dietary habits is indeed a novel and interesting perspective that has not been widely explored.

I appreciate your recognition of the methodology employed in this study, including the use of metagenomic analysis and the Comprehensive Antibiotic Resistance Database (CARD) for ARG annotation, as well as the examination of changes in ARGs based on bamboo consumption levels. The insights gained regarding the impact of giant panda diets on ARGs provide valuable knowledge for managing this issue. Furthermore, the study's findings may serve as a foundation for further research in similar areas and potentially extend to other species with different diets.

In conclusion, I appreciate your acknowledgement of the manuscript's contribution to the field of antibiotic resistance research, particularly in uncovering unconventional carriers for ARG transmission. The study provides a sound methodology and interesting results. I am grateful for the opportunity to present this unique perspective and for the thorough preparation of the manuscript. Once again, thank you for your insightful review.

Sincerely,

Hairui Wang

Reviewer 3 Report

Comments for the author of Veterinary Sciences manuscript vetsci-2616862:

The author of the Veterinary Sciences manuscript “The Impact of Bamboo Consumption on the Spread of Antibiotic Resistance genes in Giant Pandas”, present their work studying antibiotic resistance genes (ARGs) and their spread in the environment.  Specifically, they are interested in seeing how consumption of bamboo by giant pandas can impact AGR spread.  Using captive giant pandas, this research team performed metagenomic analysis and the Comprehensive Antibiotic Resistance Database (CARD) database to annotate ARGs and differentiate from the gut microbiota.  Bamboo part consumption was separated into shoots, leaves, and culms.  Their findings show that bamboo leaf consumption was associated with the highest number of ARGs while consumption of shoots affected the variety of ARGs in samples collected.  Specifically, E. coli was higher in those that consumed leaves while Klebsiella, Enterobacter, and Raoultella were higher in the shoot group.  The authors assume that ARG risk originates from soil and environmental pollution and recommend handling giant panda feces properly to monitor antimicrobial and virulence genes in gut microbiota.  This is an interesting manuscript that ties together information from captive and wild panda microbiota analysis and evaluation of ARG presence in different parts of the bamboo plant.  Since giant pandas feed on different parts of the bamboo plant at different times, this allows for the study of ARG acquisition from a single plant source.  The observational study within a unique population is clearly presented to the reader.  The data are discussed thoroughly in a manner that provides appropriate context to the detection of different bacteria and their roles in the host.  I only have a few minor comments for the authors to address.

General comments:

  1. The word CARD is used in the Abstract without spelling this out. 
  2. Some bacterial names are not italicized in the Abstract.
  3. Some of the numbers in Figure 2B do not appear to add up to those reported in the text (lines 104-108). 

Author Response

Dear Reviewer,

Thank you for your valuable feedback on our manuscript. We have taken your suggestions into consideration and made the necessary updates to improve the clarity and accuracy of our study.

Specifically, we have added a detailed explanation of CARD (Comprehensive Antibiotic Resistance Database) to provide readers with a better understanding of this critical resource. Additionally, we have italicized Klebsiella and Enterobacter in the abstract to emphasize their importance in our study.

Furthermore, we have corrected the description of the numbers presented in Figure 2B for greater accuracy and clarity.

We appreciate your insightful suggestions and believe that these revisions have significantly strengthened our manuscript. Thank you again for your time and effort in reviewing our work.

Sincerely,

Hairui Wang